# Insertion of 643bp Retrotransposon Upstream of *PPARγ* CDS Is Associated with Backfat of Large White Pigs

**DOI:** 10.3390/ani13142355

**Published:** 2023-07-19

**Authors:** Jia He, Miao Yu, Chenglin Chi, Zhanyu Du, Yao Zheng, Cai Chen, Ali Shoaib Moawad, Chengyi Song, Xiaoyan Wang

**Affiliations:** 1College of Animal Science and Technology, Yangzhou University, Yangzhou 225009, China; mx120210853@yzu.edu.cn (J.H.); mz120221567@stu.yzu.edu.cn (M.Y.); mz120191039@yzu.edu.cn (C.C.); dx120190120@yzu.edu.cn (Z.D.); mz120180996@yzu.edu.cn (Y.Z.); 007302@yzu.edu.cn (C.C.); ali.shoeib@agr.kfs.edu.eg (A.S.M.); cysong@yzu.edu.cn (C.S.); 2Department of Animal Production, Faculty of Agriculture, Kafrelsheikh University, Kafrelsheikh 33516, Egypt; 3International Joint Research Laboratory in Universities of Jiangsu Province of China for Domestic Animal Germplasm Resources and Genetic Improvement, Yangzhou 225009, China

**Keywords:** *PPARγ* gene, RIPs, repressor, molecular marker

## Abstract

**Simple Summary:**

Lipid metabolism has a substantial impact on the quality of meat in swine farming. *PPARγ* has a close association with lipid metabolism and displays high expression in adipose tissues. It is of paramount importance to control lipid metabolism to regulate *PPARγ* expression and activity in pigs. In this study, the association between the expression of the *PPARγ* gene and the backfat thickness of pigs was investigated. In recent years, retrotransposon insertion polymorphisms (RIPs), a new type of molecular marker, have emerged as a potentially valuable tool for genetic breeding in livestock and poultry. In this study, a combined retrotransposon insertion in the *PPARγ* gene of pigs was discovered that could be used as an efficient marker in selecting pigs for growth rate and lean percentage.

**Abstract:**

*PPARs* are essential regulators of mammalian fatty acid and lipid metabolism. Although the effects of genetic variations, including single nucleotide polymorphisms (SNPs) in *PPARs* genes on the phenotype of domestic animals have been investigated, there is limited information on the impact of retrotransposon insertion polymorphisms (RIPs). In this study, a combined comparative genome and polymerase chain reaction (PCR) was used to excavate the RIPs in porcine *PPARs*. We also investigated the potential effects of retrotransposon insertion on phenotype and expression patterns. This study identified the two RIPs in *PPARs* genes, namely an ERV in intron 1 of *PPARα* and a combined retrotransposon in intron 2 of *PPARγ*, designated as *PPARα*-ERV-RIP and *PPARγ*-COM-RIP, respectively. These RIPs exhibited different distribution patterns among Chinese indigenous breeds and Western commercial breeds. Individuals with the *PPARα*-ERV-RIP^+/+^ genotype (+/+ indicated homozygous with insertion) among Large White pigs had significantly higher (*p* < 0.05) corrected backfat thickness compared to those with the other two genotypes. Similarly, those with the *PPARγ*-COM-RIP^−/−^ genotype had significantly higher (*p* < 0.05) corrected backfat thickness than those with the other two genotypes in Large White pigs. Moreover, in 30-day-old Sujiang piglets, the *PPARγ* gene expression in the backfat of those with the *PPARγ*-COM-RIP^−/−^ genotype (−/− indicated homozygous without insertion) was significantly greater (*p* < 0.01) than those with other genotypes. The dual luciferase reporter gene assay demonstrated that the combined retrotransposon insertion significantly reduced the activity of the MYC promoter in both C2C12 and 3T3-L1 cells (*p* < 0.01). Therefore, the combined retrotransposon insertion could function as a repressor to decrease the expression of *PPARγ*, making *PPARγ*-COM-RIP a valuable molecular marker for assisted selection of backfat thickness in pig breeding.

## 1. Introduction

Peroxisome proliferator-activated receptors (*PPARs*) were discovered in 1990 and consist of three members: *PPARα*, *PPARγ*, and *PPARβ* [1,2]. *PPARs* acquired transcriptional activity by forming a heterodimer with the Retinoid X receptor (RXR), after which they binded to the Peroxisome proliferator response element (PPRE) to regulate the transcription of target genes [3,4,5,6]. *PPARα* was predominantly found in the liver, kidney, heart, and muscle, with a high capacity for catabolizing fatty acids [7,8]. *PPARβ* exhibited ubiquitous expression across multiple tissues [7]. *PPARγ* exhibited high expression in adipose tissue, and was also present in multiple other tissues, such as the mammary gland [9]. *PPARs* played crucial roles in metabolic syndrome [10], glucose and lipid metabolism [11], angiogenesis [12], immune responses [13], and inflammation control [14]. Multiple signaling pathways, including AMPK, were proved to be regulated by *PPARs*. Due to their vital role in regulating fatty acid and lipid metabolism, the nucleotide variations of *PPARs* genes that affect the economic traits of domesticated animals have received significant attention in recent years.

Single nucleotide polymorphisms (SNPs) in *PPARs* genes have been linked to growth, development, and meat quality in various species such as cattle and sheep [15,16,17]. A 636A>G SNP in the *PPARα* gene influenced adipose accumulation in Polish landrace pigs [18]. SNPs in the 5′ regulatory region of the *PPARβ* gene significantly impacted pig fat deposition traits [19]. Polymorphisms within the *PPARγ* promoter had a significant impact on intramuscular fat (IMF) content in the longissimus dorsi (LD) muscle of Erhualian pigs [20]. However, few research papers focus on the retrotransposon insertion polymorphism (RIPs) of these genes.

Retrotransposons are widely distributed in mammalian genomes, and their potential for mobilization can have significant impacts on the structure and function of these genomes [21]. Retrotransposon can be classified into long terminal repeat (LTR) elements, which mainly contain endogenous retrovirus (ERV) and non-LTR elements (Long interspersed nuclear elements, LINEs; Short interspersed nuclear elements, SINEs). Retrotransposons contribute to genetic diversity [22,23,24] and generate rich polymorphism [25,26,27] in the mammal genome. By changing gene expression patterns, retrotransposon insertion can result in human neurologic and psychiatric disorders [28,29,30], genetic disorders [31,32], and cancer [33,34]. RIPs have been widely used in evaluating genetic diversity [35,36], phylogenetic relationships [37], crop evolution [38], and germplasm resource analysis [39] of plants.

Previous studies have investigated RIPs in the pig genome [40]. RIPs were utilized to evaluate the genetic variation and population structure of different pig breeds [41,42]. Additionally, RIPs in important protein-coding genes were conducted to determine their association with porcine phenotypes, including coat color [43], reproductive traits [25,44], growth traits [45,46], and immunity [47]. Based on the genetic variation and population structure evaluations conducted in previous studies, as well as the importance of *PPARs* in lipid metabolism, RIPs in these genes were identified in this study. The study investigated the correlation between these RIPs and several of the pigs’ economic performances.

## 2. Materials and Methods

### 2.1. Ethical Statement

The collection of biological samples and experimental procedures involved in this study were approved by the Animal Experiment Ethics Committee of Yangzhou University (No. NSFC2020-dkxy-02, 27 March 2020).

### 2.2. Animals and Extraction of DNA and RNA

DNA extractions were conducted on multiple pig breeds, including Duroc, Landrace, Large White, Sujiang, Sushan, Erhualian, Meishan, Bama, Banna, Wuzhishan, Tibetan, and Wild boars, in order to investigate retrotransposon polymorphisms using PCR analysis. Subsequently, the extracted DNA was mixed into DNA pools. A total of eight breeds of pigs, including two commercial lean-type breeds (Large White and Duroc), two crossbreeds (Sushang and Sujiang), and four Chinese fat-type indigenous breeds (Erhualian, Jiangquhai, Fengjing, and Meishan), were genotyped for population genetic polymorphism using DNA extracted from their ear tissues. The date of age at 30 kg bodyweight, age at 100 kg bodyweight, and corrected thickness of backfat in Large White pigs were collected to analyze the correlation between RIPs and growth traits. Total RNA was prepared from tissues of Sujiang piglets. Appendix A showed the origin and number of breeds studied.

### 2.3. Extraction of DNA and RNA

DNA extraction was conducted on the ear tissues of each individual pig using the TIANamp Genomic DNA Kit (Tiangen, Beijing, China). After extraction, the concentration of each DNA sample was measured using the NanoPhotometer N60 Touch spectrophotometer (NanoPhotometer N60 Touch, Implen Gmbh, Munich, Germany) and then diluted to a concentration of 40 ng/μL. Total RNA from the liver, backfat, longissimus dorsi, and leaf fat of fifteen Sujiang piglets was extracted using TRIzol (Takara, Tokyo, Japan) to investigate the expression pattern of *PPARs*. Storing total RNA at −80 °C after measuring the concentration. 

### 2.4. RIPs Detection

The *PPARα* (NC_010447.5), *PPARβ* (NC_010449.5), and *PPARγ* (NC_010455.5) genes, along with their flanking regions (2 kb upstream and 2 kb downstream), were obtained from a reference genome of Duroc (https://ncbi.nlm.nih.gov/, accessed on 20 July 2022). The upstream 2 kb and downstream 2 kb of *PPARs* gene sequences were used as seeds to search similar sequences of other breeds or populations in the whole-genome sequencing (WGS) database (https://blast.ncbi.nlm.nih.gov/Blast.cgi, accessed on 20 July 2022) and then the corresponding sequences of other thirteen non-reference genomes were downloaded based on the beginning and ending location of comparison results. *PPARs* gene sequences information of other pig breeds or populations are provided in Appendix A. To identify large structural variants exceeding 50 bp, a ClustalX tool was used to perform a multiple sequence alignment of the gene sequences from different breeds. RepeatMasker was utilized to annotate retrotransposon insertions, including LINEs, SINEs, and ERVs, among the predicted structural variations. The identified retrotransposon insertions were referred to as RIPs and used for further identification via PCR amplification (Vazyme, Nanjing, China). Relevant primer sequences are provided in Appendix A. 

### 2.5. Population Diversity Based on RIPs

Eight pig breeds (Large White, Duroc, Sushang, Sujiang, Erhualian, Jiangquhai, Fengjing, and Meishan) were used to illustrate the genetic diversity. Genotype frequency, allele frequencies, and Hardy–Weinberg equilibrium were evaluated using the Popgene. The polymorphic information content (PIC) was calculated using the following formula:$$PIC=1-\sum_{i=1}^{n}{\left(P_{i}\right)}^{2}-\sum_{i=1}^{n-1}\sum_{j=i+1}^{n}2P_{i}^{2}P_{j}^{2}$$

### 2.6. qPCR

A total of twenty-one 30-day-old Sujiang piglets were genotyped. RNA was extracted from tissues of fifteen 30-day-old Sujiang piglets, with five individuals per genotype, and reverse transcribed to synthesize cDNA using a FastKing RT Kit (With gDNase) (TIANGEN, Beijing, China). Next, qPCR was conducted on the 7900 HT Fast Real-Time PCR System (Applied Biosystems, New York, NY, USA). The total reaction system includes 10 μL SYBR mix (Vazyme, Nanjing, China), 0.4 μL upstream and downstream primers each, 1 μL cDNA sample and 8.2 μL ddH_2_O. Gene expression was normalized to ACTB and measured using the 2^−∆∆Ct^ method. The primer sequences also are shown in Appendix A.

### 2.7. Dual-Luciferase Reporter Assay

A 643 bp insertion fragment of *PPARγ* was cloned using the Duroc genome and confirmed by sequencing. Subsequently, it was inserted into pGL3-Oct4-basic and pGL3-MYC-basic vectors to evaluate its potential as an enhancer or repressor. C2C12 and 3T3-L1 cells (1.5 × 10^5^ cells each) were cultured in 6-well plates until they reached approximately 80% cell density. Then, Luciferase reporter vectors were transfected into the cells using Lipofectamine 3000 reagents (Invitrogen, Carlsbad, CA, USA). Measurement of Luciferase activity was undertaken 48 h later using the dual luciferase reporter system (Promega, Madison, WI, USA).

### 2.8. Statistical Analysis

Using the mean ± standard deviation (SD), the data were summarized. Statistical analyses were conducted by one-way ANOVA, followed by Tukey’s post hoc test using SPSS 17.0 (SPSS, Chicago, IL, USA). The age at 100 kg body weight was adjusted based on the formula recommended by the National Swine Genetic Assessment Scheme. ANOVA was used to investigate the relationship between genotype and phenotype, focusing specifically on the age at 30 kg body weight, age at 100 kg body weight, and the corrected thickness of backfat.

## 3. Results

### 3.1. Two RIPs Generated by Retrotransposon Insertions in Pig PPARs Genes

ClustalX was utilized to analyze one reference genome and thirteen assembled non-reference genomes of *PPARs* genes, resulting in the prediction of 58 large structural variations (more than 50 bp). They were further annotated using the RepeatMasker program and 13 RIPs were predicted (Appendix A). Two RIPs were identified by PCR from the DNA pool (Figure 1A, the original electrophoretic figure is in Appendix A) and subsequently sequenced by Beijing Tsingke Biotech. Each gene was characterized by three genotypes: homozygous with insertion, heterozygous with insertion, and homozygous without insertion. These genotypes were abbreviated as +/+, +/−, and −/−. The ERV insertion identified at positions 3316558–3316851 bp on chromosome 5 of the porcine reference genome (Sscrofa11.1) in the *PPARα* gene was found to be from the subfamily ERV14-I with a length of 294 bp. A combined retrotransposon consisting of one SINEB1 (64 bp), one ERV7-I (26 bp), and one SINEA4 (197 bp) was detected in the *PPARγ* gene. The insertion was located on chromosome 13 of the porcine reference genome (Sscrofa11.1), with positions 68351107–68351108 bp. The two insertions have been designated as *PPARα*-ERV-RIP and *PPARγ*-COM-RIP. The schematic diagram of the retrotransposon insertions in *PPARα* and *PPARγ* genes are shown in Figure 1B,C.

### 3.2. Genetic Diversity of Two RIPs in Different Pig Breeds

The distribution of retrotransposon insertion was evaluated in eight different pig breeds. Additionally, a Hardy–Weinberg test was performed, and the PIC value was calculated. In Table 1, the frequency of *PPARα*-ERV-RIP^+^ was higher in crossbreed pigs (Sushan and Sujiang) and indigenous Chinese fat-type breeds (Erhualian, Jiangquhai, Fengjing, and Meishan) relative to commercial pigs. *PPARγ*-COM-RIP^−^ frequency was dominant in every pig breed except Meishan. Erhualian, one of the Chinese fat-type indigenous breeds, did not adhere to the Hardy-Weinberg equilibrium in the *PPARα*-ERV-RIP. Additionally, two commercial lean-type breeds (Large White and Duroc) and one crossbreed (Sujiang) diverged from the Hardy–Weinberg equilibrium in both RIPs. This observation suggests that these breeds may have undergone intensive selection which may have impacted the distribution of the two RIPs. The majority of the breeds displayed low to medium polymorphism, as indicated by the PIC values. However, one crossbreed (Sujiang) showed high polymorphism in the *PPARα*-ERV-RIP.

### 3.3. Correlation Analysis between RIPs and Phenotype of Large White

The relationship between two RIPs and the phenotype of Large White pigs was analyzed. As shown in Table 2, the corrected backfat thickness of *PPARα*-ERV-RIP^+/+^ individuals was significantly higher (*p* < 0.05) than that of *PPARα*-ERV-RIP^+/−^ individuals and extremely significantly higher (*p* < 0.01) than that of *PPARα*-ERV-RIP^−/−^ individuals. The age at 30 kg body weight and 100 kg body weight of *PPARα*-ERV-RIP^+/+^ individuals were significantly lower (*p* < 0.01) than that of individuals with the *PPARα*-ERV-RIP^+/−^ and *PPARα*-ERV-RIP^−/−^ genotypes. The corrected backfat thickness of *PPARγ*-COM-RIP^−/−^ individuals was significantly higher (*p* < 0.05) than that of individuals with the *PPARγ*-COM-RIP^+/+^ and *PPARγ*-COM-RIP^+/−^ genotypes. The age of 30 kg body weight of *PPARγ*-COM-RIP^−/−^ individuals was significantly lower (*p* < 0.05) than that of *PPARγ*-COM-RIP^+/−^ individuals.

### 3.4. Expression Pattern of PPARα and PPARγ in Tissues of Sujiang Pigs

The expression patterns of *PPARα* and *PPARγ* were investigated in 30-day-old Sujiang piglets with different genotypes in the liver, backfat, longissimus dorsi, and leaf fat. The results are shown in Figure 2. Only two genotypes, *PPARα*-ERV-RIP^+/+^ and *PPARα*-ERV-RIP^+/−^, were observed in *PPARα*. No significant differences in the expression of *PPARα* were observed between the two genotypes in all four tissues. In the backfat tissue, individuals with the *PPARγ*-COM-RIP^−/−^ genotype showed significantly higher (*p* < 0.01) expression of *PPARγ* as compared to those with *PPARγ*-COM-RIP^+/+^ and *PPARγ*-COM-RIP^+/−^ genotypes. Individuals with the *PPARγ*-COM-RIP^+/−^ genotype displayed significantly lower (*p* < 0.05) expression of *PPARγ* in the leaf fat compared to those with the other two genotypes.

**Table 1 animals-13-02355-t001:** Analysis of RIPs’ distribution in different breeds.

Loci	Breeds	Number	Genotype Frequency			Allele Frequency		Hardy–Weinberg Equilibrium	Polymorphic Information Content
			+/+	+/−	−/−	+	−		
*PPARα*-ERV-RIP	Large White	551	19.24	71.14	8.89	54.81	44.46	<0.01	0.38
	Duroc	24	8.33	91.67	0.00	54.17	45.83	<0.01	0.37
	Sushsan	24	79.17	20.83	0.00	89.58	10.42	0.5689	0.17
	Sujiang	21	38.10	61.90	0.00	69.05	30.95	0.0400	0.55
	Erhualian	24	25.00	75.00	0.00	62.50	37.50	0.0033	0.36
	Jiangquhai	24	95.83	4.17	0.00	97.92	2.08	0.9170	0.04
	Fengjing	24	50.00	45.83	4.17	72.92	27.08	0.4319	0.32
	Meishan	24	83.33	16.67	0.00	91.67	8.33	0.6561	0.03
*PPARγ*-COM-RIP	Large White	505	9.50	67.92	22.57	43.47	56.53	<0.01	0.37
	Duroc	24	0.00	75.00	25.00	37.50	62.50	<0.01	0.36
	Sushan	32	3.13	18.75	78.13	12.50	87.50	0.4190	0.19
	Sujiang	23	4.55	68.18	27.2	38.64	61.36	0.0400	0.37
	Erhualian	24	4.17	62.50	33.33	35.42	64.58	0.0728	0.35
	Jiangquhai	21	0.00	33.33	66.67	16.67	83.33	0.3594	0.24
	Fengjing	24	0.00	0.00	100.00	0.00	100.00	/	0.00
	Meishan	22	30.43	47.83	21.74	54.35	45.65	0.8622	0.36

**Table 2 animals-13-02355-t002:** Association analysis between *PPARs-ERV-COM-RIP* and growth traits of Large White pigs.

Genotype	Number	Age at 30 kg Body Weight/d	Age at 100 kg Body Weight/d	Correcting Backfat Thickness/mm
*PPARα*-ERV-RIP^+/+^	122	72.53 ± 9.72 ^a^	159.75 ± 5.85 ^a^	11.33 ± 2.54 ^a^
*PPARα*-ERV-RIP^+/−^	321	75.66 ± 7.82 ^b,^*	163.76 ± 9.04 ^b,^*	10.65 ± 2.65 ^b^
*PPARα*-ERV-RIP^−/−^	102	75.88 ± 6.96 ^b,^*	162.95 ± 7.37 ^b,^*	10.42 ± 2.43 ^b,^*
*PPARγ*-COM-RIP^+/+^	86	74.65 ± 9.51 ^a^	162.47 ± 8.59	10.46 ± 2.33 ^a^
*PPARγ*-COM-RIP^+/−^	304	75.73 ± 8.01 ^a^	163.19 ± 8.63	10.64 ± 2.73 ^a^
*PPARγ*-COM-RIP^−/−^	110	73.92 ± 7.22 ^b^	162.68 ± 7.65	11.17 ± 2.36 ^b^

Note: Different superscript letters indicated difference between groups (*p* < 0.05). Different superscript letters with * indicated significant difference between groups (*p* < 0.01).

### 3.5. The 643 bp Insertion from PPARγ May Act as Repressor to Regulate the MYC Promoter

The structural variation of 643 bp was analyzed using the RepeatMasker annotation information. The structural variation includes three retrotransposons: SINEB1, ERV7-1, and SINEA4. Based on the expression differences of *PPARγ* in three genotypes of Sujiang piglets, this study investigated the potential functions of this 643 bp structural variation. Two predicted promoters were 8452 bp and 10952 bp away from the transcription start site (ATG) of *PPARγ* (Figure 3A). The 643 bp insertion was cloned by PCR using the DNA template of Duroc homozygote and inserted into pGL3-Oct4-basic and pGL3-MYC-basic vectors. The recombinant plasmids were named *PPARγ*-COM^+^-Oct4-Luc^+^ and *PPARγ*-COM^+^-MYC-Luc^+^. C2C12 and 3T3-L1 cells were transfected with these two vectors. The dual luciferase reporter assay demonstrated that the retrotransposon insertion significantly suppressed MYC promoter activity in C2C12 and 3T3-L1 cells (*p* < 0.01). In contrast, in 3T3-L1 cells, the inserted fragment increased Oct4 promoter activity (*p* < 0.01). The data suggest that the inserted fragment may act as a repressor to regulate *PPARγ* gene expression. 

## 4. Discussion

Peroxisome proliferator-activated receptors (*PPARs*) are important factors in regulating lipid metabolism, adipogenesis, inflammatory response, and cell differentiation [48]. *PPARα* and *PPARγ*, subtypes of the *PPARs* gene, played an important role in fatty acid catabolism and adipogenesis [3,4,5,6]. This study found 13 putative polymorphic loci in porcine *PPARs* genes resulting from retrotransposon insertion. *PPARα*-ERV-RIP and *PPARγ*-COM-RIP were identified by PCR, both of which were located in the intron. The introns of genes in higher eukaryotes contain a large number of regulatory elements [49,50,51]. Approximately 90% of human and mouse genes included TEs in their introns [52]. In the pig genome, retrotransposons that were inserted into the introns of protein-coding genes accounted for a proportion of 35.1 [53]. Massive genomic variation may result from the existence of transposable elements (TEs) [54,55].

The distribution of two RIPs was evaluated in the different types of pig breeds, including fat-type, lean-type, and crossbreed. *PPARα*-ERV-RIP^+^ was dominant in the crossbreeds and fat-type, while *PPARγ*-COM-RIP^−^ was dominant in the lean-type. Commercial pigs have a higher lean percentage and thinner backfat than Chinese native pigs [56,57,58,59]. This may be attributed to artificial and natural selection. Compared to other animals, pigs are more efficient in adipogenesis and fat deposition. In recent years, consumers prefer pork with a high lean percentage and commercial pigs were more popular. Therefore, the selection of lean percentage in pigs may result in the nucleotide changes of genes related to fat deposition such as *PPARs*. In commercial lean-type (Large White and Duroc) and crossbreed (Sujiang), which have diverged from the Hardy–Weinberg equilibrium, a higher frequency of the heterozygous with insertion (+/−) was observed. This may be due to the fact that these breeds may have undergone strong selection pressure for lean meat percentage. Additionally, in Erhualian pigs at the *PPARγ*-COM-RIP, the genotype frequency of heterozygous with insertion (+/−) was also higher than that of the other two genotypes. The Erhualian pigs were kept in breed conservation and whether this attribute affects selection should be further studied.

The association between these two RIPs and the growth traits of Large White pigs was investigated. The results showed that *PPARα*-ERV-RIP^+/+^ individuals grew more quickly and had thicker backfat (*p* < 0.05), while *PPARγ*-COM-RIP^+/+^ individuals grew more slowly and had thinner backfat (*p* < 0.05), which may indicate that the ERV and SINE insertion in *PPARs* may affect the porcine phenotype and the mechanism should be further studied. Porcine backfat thickness is one of the important factors affecting pig lean percentage. Transposons can affect phenotypes by regulating gene expression. Laiwu pigs have a thicker backfat and higher expression of *PPARγ* in backfat than western pig breeds [60]. In tissues of Sujiang piglets, *PPARα* expression was not significantly different between *PPARα*-ERV-RIP^+/+^ and *PPARα*-ERV-RIP^+/−^ genotypes individuals. This might be related to differences in gene expression in different tissues and developmental stages [61]. In the backfat, *PPARγ* expression in the COM^−/−^ genotype was much higher than that of the other two genotypes (*p* < 0.05). Experimental results about the expression of *PPARγ* indicated that the combined retrotransposon insertion might reduce the expression of *PPARγ* in the backfat. However, the homozygous without ERV insertion of 30-day-old Sujiang piglets was not found and the expression difference of *PPARα* between the homozygous with and without ERV insertion could not be observed. So, the mechanism of how the ERV insertion effected the phenotype of Large White pigs should be further studied.

A dual luciferase reporter assay was subsequently conducted to verify the function of combined retrotransposon insertion. The combined retrotransposon insertion significantly decreased the MYC promoter activity in both C2C12 and 3T3-L1 cells. However, the insertion resulted in increased Oct4 promoter activity specifically in 3T3-L1 cells. In our previous studies, the SINE retrotransposon insertion in the *GHR* gene of the pig could lead to a reduction in *GHR* gene expression [45]. Additionally, SINE insertion could also function as an enhancer, increasing the expression of *BMPR1B* in the ovaries and influencing the reproductive traits of Large White pigs [44]. The ERV insertion may act as an enhancer affecting the regulation of the *TLR* signaling pathways [47]. In this paper, the combined retrotransposon insertion acted as a repressor to regulate the MYC promoter activity in C2C12 and 3T3-L1 cells. Previous studies have confirmed that retrotransposon insertions could perform different functions. This difference may be related to different promoters [62], different cells [63], or methylation status [64]. The reason the insertion did not reduce Oct4 promoter activity in C2C12 cells may be due to the fact that the combined retrotransposon could not work with the Oct4 promoter in C2C12 cells. Since the combined retrotransposon insertion was present upstream of the CDS domain, combined with the results of the expression pattern and growth trait affected by the insertion in *PPARγ,* the combined retrotransposon could serve as a repressor in regulating the expression of *PPARγ* and affect the fat deposition in porcine backfat. In the human genome, SINE-VNTR-*Alu* (SVA), a young composite retrotransposon, was presented in about 2700 copies [65]. SVA might impact the genome and sometimes cause disease [66]. In the *LRIG2* gene, SVA, as methylation quantitative trait loci (mQTL), could regulate the expression levels of *LRIG2* [67]. Therefore, multiple transposons could combine to affect the architecture of the host animal genome. However, in this study, the combined retrotransposon insertion existed only in one copy. Further research is needed to reveal the possible mechanism of formation of this insertion and the function of each retrotransposon. 

## 5. Conclusions

This study identified two RIPs, one in intron 1 of *PPARα* and another in intron 2 of *PPARγ*, through comparative genomics and PCR verification. The distribution of *PPARα*-ERV-RIP^+^ and *PPARγ*-COM-RIP^+^ was different between commercial pigs and Chinese local pigs. These two RIPs displayed a significant correlation in growth rate and backfat thickness in Large White pigs. Furthermore, the combined retrotransponson insertion reduced the *PPARγ* expression in the back fat of Sujiang piglets and decreased the MYC promoter activity in C2C12 and 3T3-L1 cells (*p <* 0.01) according to a dual luciferase reporter gene assay. Therefore, *PPARγ*-COM-RIP could serve as useful markers in selecting pigs aimed at growth rate and lean percentage and the insertion may affect the host gene expression and further affect the fat deposition.

## Figures and Tables

**Figure 1 animals-13-02355-f001:**
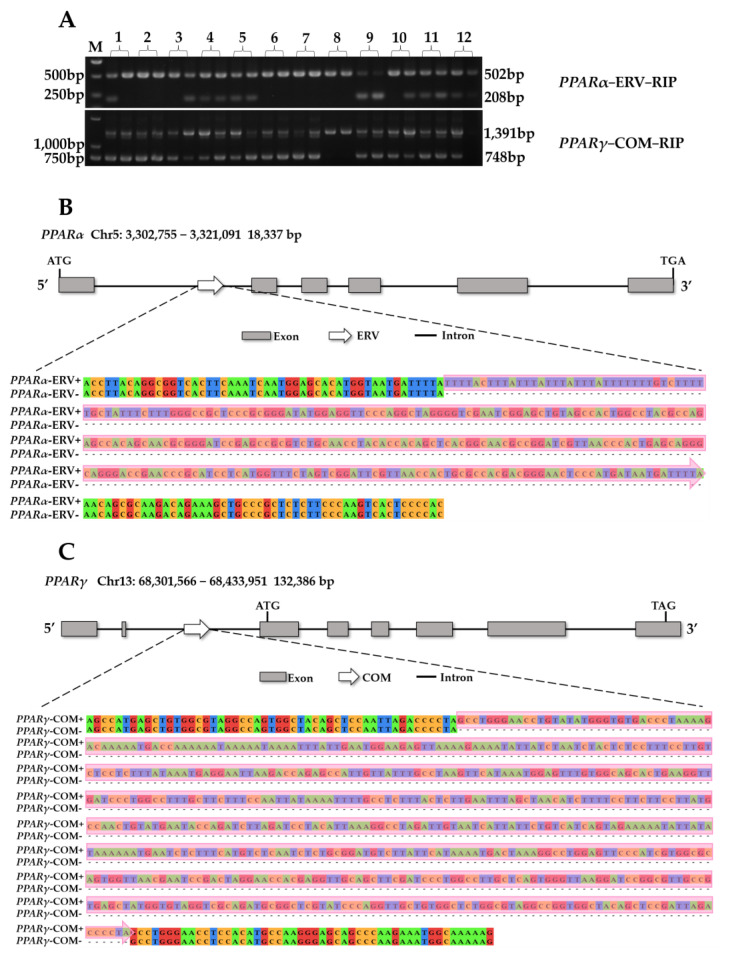
RIPs identification in *PPARs* genes. (**A**) PCR detection of RIPs and genotype schematic. 1. Duroc, 2. Landrace, 3. Large White, 4. Sujiang, 5. Sushan, 6. Erhualian, 7. Meishan, 8. Bama, 9. Banna, 10. Wuzhishan, 11. Tibetan, 12. Wild boars. M: DNA marker DL2000. The electrophoretic results showed that the homozygous with insertion (+/+) exhibited a single long band, the heterozygous with insertion (+/−) exhibited both long and short bands, and the homozygous without insertion (−/−) exhibited a single short band. (**B**) The ERV sequence in *PPARα*-ERV-RIP site with and without ERV insertion and its location on *PPARα* gene. (**C**) The sequence in *PPARγ*-COM-RIP site with and without combined insertion and its location on *PPARγ* gene.

**Figure 2 animals-13-02355-f002:**
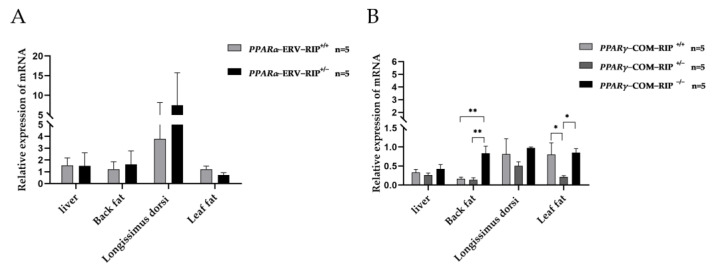
The expression pattern of two loci (*PPARα*-ERV-RIP and *PPARγ*-COM-RIP) in tissues of 30-day-old Sujiang piglets. (**A**) Relative expression of *PPARα* gene in Sujiang piglets’ tissues. (**B**) Relative expression of *PPARγ* gene in Sujiang piglets’ tissues. The values are shown as mean ± SD, and * shows *p* < 0.05, ** shows *p* < 0.01.

**Figure 3 animals-13-02355-f003:**
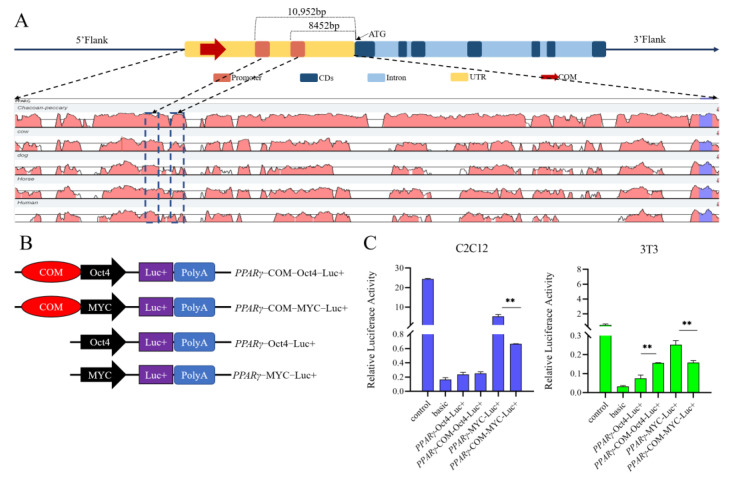
A combined retrotransposon insertion served as a regulating element to affect the promoter activities. (**A**) Schematic diagram of promoter prediction. (**B**) Schematic diagram of the recombinant vector structure. (**C**) Impact of combined insertion of *PPARγ*-COM-RIP on the promoter activity of MYC and Oct4 in C2C12 and 3T3-L1 cells by dual-luciferase reporter assay. ** shows *p* < 0.01.

## Data Availability

All data needed to evaluate the conclusions in this paper are present either in the main text or the Appendix A.

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
