# Peer review of "Insertion of 643bp Retrotransposon Upstream of PPARγ CDS Is Associated with Backfat of Large White Pigs"

_animals, 2023, doi:10.3390/ani13142355_

Round 1
Reviewer 1 Report
In this study, the authors confirmed two retrotransposon insertion mutations in PPARγ gene. Also, the authors confirmed that the of 643bp Retrotransposon upstream of PPARγ CDS is associated with backfat of Large White pig. This study offered new information for the study of retrotransposon mutations in pigs. Further,it offered new markers for pig breeding.
Minor comments
1.Line 17: two full stops.
2.Line 34-37: The result is in which breeds? Large white pigs?
3.Line 128: which eight pig breeds were used to illustrate genetic diversity? Why did you choose them?
4.Line 133: please give the total size of the piglets.
5.Line 197: Were correlation analysis done for other breeds? Did they show similar results?
6.Line 209: as the title showed, this paper is mainly about large white pigs, but the expression pattern is validated in tissues of Sujiang pigs, can the results of it represent that of large white pigs?
7.Figure 2: please enlarge figure 2 and superscripts
8.Figure 4: enlarge figure 4C
9.Line 275-276: change “showed that” to “about”
Reviewer 2 Report
The manuscript of Jia Het et al. reports the analysis of two retrotransposition insertion polymorphism (RIP) in the PPARy gene in pigs. It is argued that genetic selection could take advantage of this polymorphism to improve growth rate and lean percentage. The manuscript is overall well constructed, and the results are well-founded. The description of the methods, the figures as well as some few English formulations could be improved. We list here some minor proposed modifications:
-
The 2.4 paragraph of the Materials and Methods does not enable to understand exactly how structural variations were identified. How was the PPAR gene genome region identified in the thirteen assemblies ? Did ClustalX provide a single multiple alignment for the complete region ? “RIP detection” would probably be a better title for this paragraph
-
The figures are not readable, in particular Fig 1C and 1D, Fig4A (base pairs and color labels), to a lesser extent Fig2. The captions do not describe adequately the figures.
-
There is no Table 1. The reader has to guess the content of Table 2 (frequencies ?, the meaning of the P-value, departure from HW equilibrium?). The unit of measurement in Table 2 is not given (month ?). “Age at 30kg body weight” is perhaps more appropriate than “Age of 30kg body weight”
-
The + and - superscript in the abstract and more generally in the paper are not easily readable, forcing the reader to guess the genotype from the sentence
-
The causes of the departure from HW equilibrium for the two commercial lean-type breeds and the cross-breed could be more thoroughly discussed
Reviewer 3 Report
In this study, the authors validated 2 RIPs by comparative genomics and PCR, and further association analysis illustrated their significant association with corrected backfat thickness in pigs. Finally, dual luciferase reporter gene analysis showed that PPARγ-COM-RIP could act as a repressor to reduce PPARγ expression. The results of this study are of guiding significance for the molecular breeding of pigs. Before acceptance some points need clarification.
L94-104: Need to specify how many samples were used to validate polymorphisms? How many samples were genotyped?
L133: Why was the age of 30 days chosen for the expression analysis?
Figure 1: Genotypes need to be labeled in Figure 1A.
L184: I did not find table 1.
L272-273: Here the authors can try to explain why the expression is not significantly different and why there is a significant correlation between growth traits in different genotypes in association analysis.
L278-279: Why did the insertion not reduce Oct4 promoter activity in C2C12 cells?
L300: There are too many results described in the conclusion, so it is suggested that the author should make a further concise and summary.
Minor editing of English language required.
